# Safety and Evaluation of the Immune Response of Coronavirus Nosode (BiosimCovex) in Healthy Volunteers: A Preliminary Study Extending the Homeopathic Pathogenetic Trial

**DOI:** 10.3390/medicines10010008

**Published:** 2022-12-30

**Authors:** Paul Herscu, Gitanjali Talele, Shashikant Vaidya, Rajesh Shah

**Affiliations:** 1Herscu Laboratory, Research Division, 356 Middle Street, Amherst, MA 01002, USA; 2Life Force Foundation Trust, 411 Krushal Commercial Complex, Chembur, Mumbai 400089, Maharashtra, India; 3Assistant Director and HOD Microbiology Department, Haffkine Institute for Training Research and Testing, Acharya Dhonde Marg, Parel Village, Parmanand Wadi, Parel, Mumbai 400012, Maharashtra, India

**Keywords:** coronavirus, SARS-CoV-2, homeopathic pathogenetic trials (HPT), COVID-19, BiosimCovex, nosode, potentization, homeopathy, safety, immune response, phase-1 study

## Abstract

**Objectives**: Regulatory clinical Phase I studies are aimed at establishing the human safety of an active pharmaceutical agent to be later marketed as a drug. Since homeopathic medicines are prepared by a potentizing method using alcohol, past a certain dilution, their toxicity/infectivity is assumed to be unlikely. We aimed to develop a bridge study between homeopathic pathogenetic trials and clinical trials. The primary purpose was to evaluate the safety of a nosode, developed from clinical samples of a COVID-19 patient. The secondary objectives were to explore whether a nosode developed for a specific clinical purpose, such as use during an epidemic, may elicit laboratory signals worthy of further exploration. **Methods**: An open-label study was designed to evaluate the safety and immune response of the Coronavirus nosode BiosimCovex, given orally on three consecutive days to ten healthy volunteers. Clinical examinations, laboratory safety and immune parameters were established. Interferon–gamma, Interleukin-6, and CD 4 were measured. (CTRI registration number: CTRI/2020/05/025496). **Results**: No serious/fatal adverse events were reported. Laboratory tests to measure safety were unchanged. Three subjects showed elevated Interleukin-6 (IL-6) on day 17 in comparison to the baseline, and ten subjects showed elevated IL-6 on day 34. A significant difference between IL-6 observations, calculated by repeated measures ANOVA, was found to be highly significant. On day 60, the IL-6 values of nine subjects were found to return to normal. Corresponding CD4 cell elevation was observed on day 60, when compared to day 34. **Conclusions**: HPT may potentially extend into physiological changes with regards to immune response and should encourage future studies.

## 1. Introduction

Clinical Phase I trials for the regulatory approval of drugs are generally small and are conducted to primarily examine the safety of new medicines in healthy subjects before proceeding further into clinical trials that modify disease. The treatment of illnesses caused by SARS-CoV-2 is always potentially limited by the toxicity of medications [1]. The continued need for supportive prophylaxis measures, immunotherapeutics and immunomodulators which may contribute to controlling pandemics; therefore, we considered the development of potentized preparations (nosodes) [2].

In the homeopathic system of medicine, the drugs are potentized via dilution and succussion, rendering the source material open to potential nanoparticles [3]. It is assumed that there is little or no toxicological risk remaining past a certain potency when consumed by human subjects. For example, the potentization process leading to a 30C potency, has led to an extreme dilution, making the concentration difficult to measure In addition, the process kills any organism in the original material (bacteria or virus, in the case of nosodes) due to the continued and repeated exposure of the organism to approximately 90% alcohol, used as a medium for dilution at every step. As a consequence, human trials for safety have typically not been a practice for new or old homeopathic drug discovery. Instead, the model of the Homeopathic Pathogenetic Trial (HPT/drug proving) was developed to understand the primary effects of the drug substance by recording symptoms during the trial, which help determine the therapeutic indications, based on the fundamental homeopathic principle of the law of Similars. Such primary effects are recorded in the source books, such as Materia Medica Pura (by Samuel Hahnemann [3]), Encyclopedia of Materia Medica (by TF Allen [4]) and Hering Guiding Symptoms (by Constantine Hering [5]).

There are nearly 1500 drug substances recorded in various homeopathic pharmacopeias (Indian, British, German, and American Pharmacopeia) [6,7] that are fully or partly ‘proved’ in human trials and have been used by practitioners the world over for the past two centuries. They are made available across homeopathic pharmacies, as over-the-counter products, or as dietary supplements, depending on national regulatory requirements. Potentized homeopathic drugs that have been in use for over one hundred years were not required to undergo any toxicity studies in animals or safety studies in humans.

As an example, ‘The Drug and Cosmetic Act’ (India) states that, to be included in the pharmacopeia, the drug must be along-used homeopathic drug whose therapeutic efficacy is recorded in data and established by long clinical use [8]. The Act does not ask for animal or human toxicity studies for homeopathic medicines.

Controversy remains surrounding nosode use as a potential disease prophylactic during epidemics. Loeb et al. [9] showed that homeopathic ‘vaccines’ do not elicit antibody responses against diphtheria, pertussis, tetanus, and MMR. If nosodes provide some level of protection during epidemics, they may do so by a mechanism different to that of conventional vaccines.

The aim of this study is to bridge the homeopathic pathogenetic trial [10] technique with the safety testing commonly found in Phase I drug trials; this study tests the new medicine BiosimCovex (Coronavirus nosode/CVN01). Secondly, since this medicine is developed for its potential use as a protective prophylaxis in the current COVID-19 pandemic, we examined whether a homeopathically-prepared medicine may elicit certain blood value changes related to immune response, which may signal potential therapeutic use. Specifically, with a need for generating scientific data, the present study does not only investigate IgG/IgM levels against the SARS-CoV-2 Spike Protein, but also includes other immune parameters that may potentially impact disease presentation. We included the study of relevant immunomodulatory effects. IL-6 promotes the specific differentiation of naive CD4 T cells, thus performing an important function in linking the innate to the acquired immune response [11].

### Objective

To evaluate the safety of BiosimCovex in healthy volunteers, this study evaluated potential immune response signals and recorded subjective symptoms elicited in the trial, historically and commonly used in homeopathic pathogenetic trials.

## 2. Materials and Methods

### 2.1. BiosimCovex (Coronavirus Nosode)

A Coronavirus nosode, initially coded as CVN01 and subsequently labeled as BiosimCovex, was prepared from a clinical sample [12] of an oropharyngeal swab of a patient; its use had been approved by the institutional Ethics and Biosafety Committee having confirmed, by RT-PCR and genome sequencing, the infection of SARS-CoV-2 at Haffkine Institute, Mumbai. This followed the nosode-making guidelines, as per the Homeopathy Pharmacopoeia of India, and the Standard Operating Procedure, approved by the Life Force Foundation Trust’s Scientific Advisory Body [13,14]. This nosode is prepared from a clinical sample, as mentioned in the N-IV group in the HPI [13]. Almost all the nosodes (such as Psorinum, Medorrhinum, Tuberculinum Carcinosin, Syphilinum) in the homeopathic literature and available on the market have been made from the clinical samples [15]. The ethical and biosafety approvals for the preparation of the nosode were obtained from the applicable institutional committees. The samples were handled, and the potencies were prepared in a Biosafety Level-2 (BSL-2) containment lab with BSL-3 practices. Standard precautions, as per biosafety requirements, were followed during the preparation process.

The clinical sample selected to prepare the nosode had the lower cycle threshold (CT) and was identified/characterized by gene sequencing. The CT value was approximately 17 and corresponded to 4–9 log10, amounting to the count of more than 20 billion. Based on the CT value of the sample, the corresponding viral copies, as per the published literature, were sufficient to start the homeopathy potencies [16]. Potencies 1C to 4C were prepared by using Water for Injection (WFI) as a vehicle. In total, 0.03 mL of OSN was taken in a suitable glass bottle, allowing one-third space for succussion and the 2.97 mL of WFI that was added to make a 3 mL volume. The bottle was given 10 strokes with the help of a hand-potentizer in a bio-safety cabinet to arrive at 1C potency. As per the method of preparation described in the HPI, the 1:99 ratio was maintained for further potencies. The nosode, in its final preparation, was delivered on a dry size 30 globule, without alcohol present. Finally, for safety, RT-PCR confirmed that no Coronavirus was detected beyond the 3C potency. To document that usable 30C potency is also devoid of any viral material, above 3C potencies were also tested by using RT-PCR.

### 2.2. Study Design

An open-label, non-randomized study was designed to examine the response of BiosimCovex administered orally to healthy volunteers. This study aimed to examine safety in terms of clinical effects and blood parameters, and to evaluate certain immune responses at the baseline and at days 17, 34, and 60.

This study was conducted at a single center in Mumbai. The first volunteer was enrolled on 10 June 2020, and the last volunteer’s visit was completed on 12 August 2020. The study protocol, amendments, and informed consent forms (ICF) were reviewed and approved by the Ethics Committee constituted by Homeopathy India Private Limited. Written informed consent was obtained from each volunteer before the performance of any study-specific procedures. This study was conducted in accordance with the guidelines of the Good Clinical Practice and the ethical principles outlined in the Declaration of Helsinki 2008.

### 2.3. Study Sample

This is a category ‘nosode’ preliminary HPT, with ten individuals proposed by the scientific advisory board as appropriate for a Phase I trial. Ten healthy volunteers (18–65 years, 4 males and 6 females), with no known major untreated diseases, with normal routine laboratory parameters during screening, were enrolled in the trial (Figure 1). The basic demographic information is presented in Table 1. Subjects who currently had or were recently diagnosed with COVID-19 infections were excluded from the study. Full inclusion and exclusion criteria included the following:

### 2.4. Inclusion Criteria

Age: 18–65 years, both males and females, healthy individuals with no major untreated diseases and normal routine laboratory parameters during screening. Participants who were informed of the nature of the study and were willing to give written informed consent were enrolled in the study.

### 2.5. Exclusion Criteria

Any subject with a known disease or condition which might compromise the general health or comorbid conditions, subjects who had been confirmed to have COVID-19 and had been isolated for its treatment, subjects who had recently suffered and recovered from COVID-19, persons with a known history of allergies and food hypersensitivity, women during pregnancy, puerperium and breast-feeding, subjects who had participated in another clinical trial during the last 6 months and with other conditions, which, in the opinion of the investigators, could make the patient unsuitable for enrolment or could interfere in their adherence to of the study protocol, were excluded from the study.

### 2.6. Study Method

Subjects were administered with six doses of nosode BiosimCovex 30C (pill size 30) (Manufactured at Haffkine’s Institute-Batch no CoviNo-052020, Mfg. date: May 2020) as six pills, twice daily, for three consecutive days. Pre and post examinations, such as physical examinations, vital signs, and laboratory investigations were taken at baseline, on days 17, 34, and 60. The participants were monitored thoroughly. Based on the data in the source documents and laboratory values, the safety results were reported. Samples were sent to the testing laboratory within 30 min of collection at ambient conditions to avoid any degradation. The central laboratory carefully processed the samples upon arrival.

### 2.7. Approvals

The trial protocol was reviewed by the Scientific Advisory Board and approved by an Ethics Committee, named ‘Institutional Ethics Committee’, letter dated 27 May 2020. The trial was registered at the Clinical Trial Registry of India (CTRI) with the trial registration number: CTRI/2020/05/025496.

### 2.8. Study End Points and Statistical Assessments

The primary endpoint was the determination of the safety of the BiosimCovex, and the secondary endpoint was the evaluation of the changes in the immune response parameters.

As the observations made after a specific time interval demonstrate the pre- and post-treatment effects, and the data is continuous in nature, the t test for Paired Two Sample for Means was explored. Statistical analyses were performed using Software SPSS (Version 1.0.0.1447 IBM Corp., Armonk, New York, United States). For a continuous dependent variable IL-6, where the sample was collected at multiple visits, a significant difference between observations in each visit was calculated by repeated measures ANOVA.

### 2.9. Safety Assessments

Safety assessments included the monitoring of subjects for any unexpected symptoms, adverse events, serious adverse events, as assessed by predefined questionnaires, clinical and laboratory investigation results, blood pressure, physical examination findings, and general well-being, as assessed by questionnaire.

### 2.10. Safety Parameters

Complete Blood Count (CBC), C-Reactive Protein (CRP), Liver Function Test (LFT), performed on the fully automated analyzer (XL-200), and Serum ferritin level, measured by CMIA, and the chemiluminescent assay, were measured at baseline and day 17. The RT-PCR tests for the laboratory verification of COVID-19 were not permitted by the government for healthy persons (due to the shortage of kits in the country during the study period). Instead, the status of the COVID-19 IgG/IgM against the SARS-CoV-2 Spike Protein, measured by enzyme-linked immunosorbent assay (ELISA) method (Index >1.1 positive, 0.9–1.1 borderline, and <0.9 negative), were investigated at baseline, days 17, 34, and 60, along with the CBC and ferritin, to confirm that the subjects were not infected.

### 2.11. Immune Response Laboratory Method Parameters

An interferon–gamma (IFN-γ) assay was conducted by the Chemiluminescent assay (CLIA) method, using a value less than 0.438 IU/ML as negative, and above that as positive; this was measured at the baseline and day 17. In addition, Interleukin-6, using a value greater than 7.00 pg/mL as positive and an electro chemiluminescent immunoassay (ECLIA, Elecsys 2021 Modular Analytics E170 Cobas) method, was measured at the baseline and days 17, 34 and 60. After initiating the study, and observing elevations of Interleukin-6, we amended the protocol to also investigate the CD4 panel, measured at days 34 and 60.

## 3. Results

There were no serious or fatal adverse events during the study. The laboratory basic biochemistry and liver function tests were not affected by BiosimCovex 30C (Table 2). The P-value results of CBC, CRP, LFT, and the Serum ferritin level for all subjects at baseline, compared to day 17, were unchanged, and in the normal reference range. The subjects were closely monitored and did not show any abnormal clinical changes. Based on these findings, no safety concerns were observed in the study population.

As this manuscript is focused on safety and potential immune response signals, symptoms that are usually cataloged during a homeopathic pathogenetic trial were cataloged and are published elsewhere.

Regarding immune response, such as cytokine change, three subjects showed elevated IL-6 on day 17 as compared to baseline, and all 10 subjects showed elevated IL-6 on day 34. With the repetition of the IL-6 measure at day 60, the values of 9 subjects were not different (returned) compared to the baseline. A significant difference between IL-6 observations, calculated by repeated measures ANOVA, was found highly significant (*p* value 0). (Table 3) The subjects did not show any clinically untoward symptoms during the time of the elevated IL-6 values. On the CD4 panel, the absolute CD4 count showed a significant elevation following the IL-6 timeframe on day 60, in comparison to day 34. (Table 4). Individual supporting data presented in Appendix A.

IFN-γ was measured at the baseline and day 17, [Baseline Mean (SD) 0.016 (0.025) and day 17 Mean (SD) 0.13 (0.12)] and COVID-19 IgG and IgM measured at the baseline (0.157/0.264), days 17 (0.352/0.207), 34 (0.220/0.231), or 60 (0.24/0.25), did not show any activity indicating active exposure to SARS-CoV-2 virus during the dates measured.

## 4. Discussion

In conventional homeopathy, one conducts human pathogenetic trials (known as drug provings), without carrying out any animal toxicity studies or safety studies, found in precedents with earlier commonly-used nosodes such as Psorinum [17], Medorrhinum [13], Tuberculinum [18], Carcinosin [19], Variolinum [13], and more.

The current trial of BiosimCovex in 30C potency studied in ten human subjects did not elicit safety signals. It remains debatable within the homeopathic community whether the conventional homeopathic pathogenetic trials should require acute, subacute, or chronic toxicity in animal models, for examining the safety of nosodes; this is due to its potentized process which kills the original germs, if diluted, succussed and immersed in 90% alcohol [20] during the process.

This human safety study was combined with the evaluation of the potential effects of the nosode on the immune response. If the nosode was to be examined particularly for its prophylactic potential during the pandemic, it is reasonable to expect an immune response during the trial, and therefore this was evaluated. Being a pleiotropic cytokine, Interleukin-6 (IL-6) is known to induce an antigen-specific immune response and inflammatory reaction, activating the host defense mechanisms performing diverse functions. Studies of genetically modified animal models suggest that IL-6 has a role in both the innate and adaptive immune responses that protect the host from a variety of infections [21].

In this study, the mild elevation of IL-6 in all the subjects may be linked to an immune response, as there were no corresponding clinical signs and symptoms of inflammatory or pathological response. In addition, there was no increase in the C-Reactive Protein, despite an increase in IL-6, suggestive of non-inflammatory and non-pathological responses, thereby suggesting an immune response [22].

IL-6 levels, increasing by several tens or even hundreds of pg/mL, are observed in chronic diseases, depending on severity. Values of more than 1000 pg/mL can occur during septic shock or a cytokine storm and in severe cases [8]. The excessive or sustained production of IL-6 is involved in various diseases [23].

The roles of IL-6 and IFN-γ are both important and well described, and both are elevated greatly in COVID-19 patients [24]. Importantly, they are both investigated as predictors of outcomes, including mortality. For example, IFN-γ is investigated as an independent risk factor associated with mortality in the moderate and severe presentation of COVID-19 [25]. Indeed, an emerging strategy is aimed directly at reversing the decline in health status and lowering mortality rates in the acutely COVID-19 ill patient [26,27].

In our studies, unlike extreme changes, the IL-6 levels increased to an average of five folds (statistically significant *p* value ≤ 0.05). The current cytokine fluctuation was observable on days 17 and 30, but not on day 60. IL-6 levels were reduced to baseline in 9 of the 10 subjects by day 60. Interestingly, the timing of elevated IL-6 found in these subjects corresponds to the commonly observed timing of symptom production during a conventional homeopathic pathogenetic trial [28].

In an influenza vaccination study, Mohanty et al. [29] had reported an increase in cytokines, including IL-6, as an immune response following vaccination. An initial mild rise in IL-6 may be considered a positive sign of immune response. They had also reported a time-dependent elevation in intracellular cytokine production in monocytes with the highest levels found at around day 28. In our study, subjects began showing an increase in IL-6 on day 17, all having a significant rise by day 34, and subsequent decline by day 60. This corresponded to an increase in CD4 count by day 60, as compared to day 34 in 7 subjects, thereby potentially suggesting the activation of innate immunity in response to the nosode.

IL-6 is considered an important mediator of the immune response, particularly by directly acting on CD4 T cells and determining their effector functions [30]. We observed IL-6 elevation with a potential increase in the CD4 helper cell count at corresponding visits, which is again an expressive marker of immunity.

Interestingly, this study observed a signal associated with the use of BiosimCovex, potentially activating the IL-6 level in all the subjects and potentially influencing the innate immune response by CD4 T cell differentiation in some. The T-cell-activation potential of the nosode needs further studies. IFN-γ and immunoglobulins were not released at any level.

While IL-6 stimulating the production of CRP [31] is well studied, we did not observe any CRP [Baseline (1.4), day 17 (1.61), and day 34 (1.9)] elevation in this study. Lymphocyte count was measured after IL-6 data demonstrated results; therefore, no baseline data is available. With Lymphocyte counts remaining in the normal range, there could be no fear that autoimmunity is triggered by this nosode.

As discussed by Pérez-Galarza, et al., there are several key roles considered for CD4 in mediating the immune response when related to COVID-19, including originating various phenotype T helper cells, such as Th1, to helping B cell create a long-lived antibody response, to helping CD8 expand the primary immune response, to stimulating natural killer cells’ cytotoxicity, to helping in viral clearance, and perhaps, memory CD4 helping to prompt immune defense systems [32].

The clinical symptoms produced by the subjects were mild, short-lasting, and self-limiting, and were carefully documented and reported in a separate homeopathic pathogenetic trial article [10]. While a larger placebo-controlled HPT could be conducted in the future to elicit more symptoms, the effects of the COVID-19 virus are well understood due to their pandemic prevalence, and it may not be imperative to conduct symptomatic HPT for its immediate prescribing as a prophylactic or therapeutic indication. Indeed, in a related study, the authors conducted a randomized double-blind, placebo-controlled feasibility study, evaluating the efficacy of homeopathic medicines in the prevention of COVID-19 in a quarantined population. In 2233 quarantined individuals who were exposed to people diagnosed with COVID-19, the subjects in the BiosimCovex arm reported signals of efficacy, with a significantly smaller number of individuals testing COVID-19 positive compared to the placebo group, as well as a shorter duration of illness compared to the placebo [33].

IL-6 and CD4 changes were mild, but signal potential should be explored in a larger follow-up trial. It is expected that these markers would show fewer changes than during an infection or when elicited by a vaccine, though if these were seen in a large trial, it could help the understanding of how homeopathic medicines might help in this public health crisis. Interestingly, this finding matches previous results from other diseases. For example, in a study on tularemia, Jonas presented results that were significantly better than the placebo (22%), but not as effective as vaccinations (100%) [34]. If there is some level of protection afforded by homeopathic prophylaxis, it may be useful for those who are unable to tolerate vaccines or do not have access to them.

As epidemic and pandemic emergencies present with time constraints, immediate prophylactic and therapeutic solutions may not wait for the lengthy process of new nosode drug discovery. It is essential to develop faster, effective, yet safer methods of therapy; these may include the development of nosodes from clinical samples or inactivated microbial components, conducting safety studies, and evaluating specific immunological tests to assess potential clinical usefulness. Should this strategy prove even partially effective, it may fill the gap that occurs from the start of an epidemic until vaccine development, production and distribution are underway. Further, with genomic variations, mutations, and adoptions [35], it is imperative to search for alternate immune preventive or therapeutic measures.

The limitations include Type 1 errors due to the small sample size in the trial and a lack of a placebo-controlled arm. The lack of a placebo arm points us to the potential association of the immune response, but not to causation. Another limitation is potentially the too few immune markers being examined. Unfortunately, we added the CD4 count after we observed a rise in IL-6, and, therefore, did not capture CD4 and Lymphocyte count baseline levels. The study result is indicative of the immune response; however, a confirmatory study should be repeated in the future in a larger sample size, with controls, and testing a wider number of immune markers.

## 5. Conclusions

No safety concerns were observed in the small study population by consuming BiosimCovex. In this noncontrolled open-label study, while antibodies to COVID-19 remained negative, a transient immune response may have been observed in elevated IL-6 in all the subjects, with a corresponding increase in the CD4 count on day 60 in comparison to day 34. A larger placebo-controlled bridge study may be possible to explore the induction of biochemical changes that may occur by ingestion of the nosode.

## Figures and Tables

**Figure 1 medicines-10-00008-f001:**
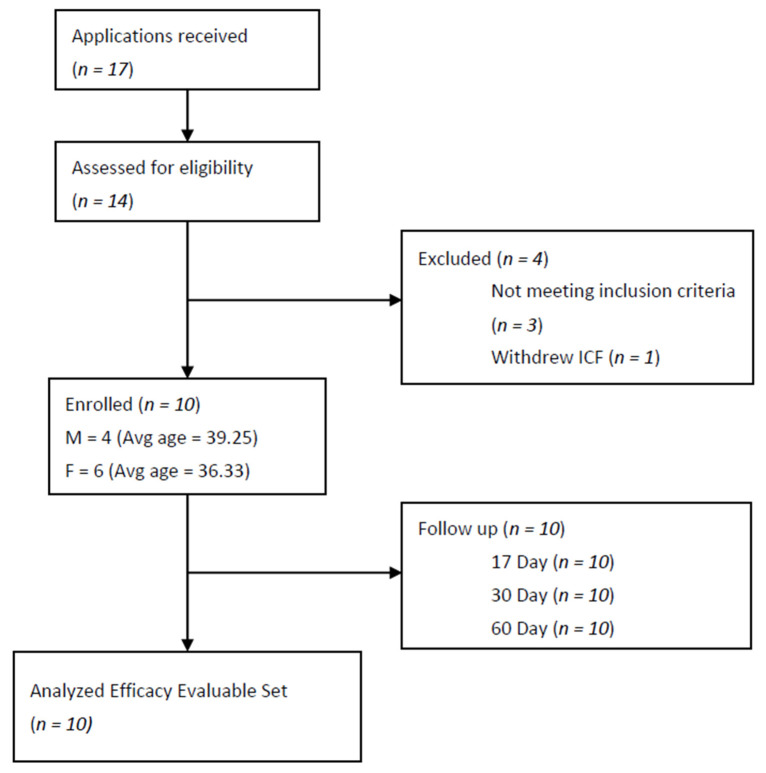
Flow chart of an open-label, homeopathic pathogenetic trial for safety and the evaluation of the immune response of BiosimCovex in healthy volunteers. ICF = Informed Consent Form.

**Table 1 medicines-10-00008-t001:** Demographics of Individual Subjects.

Subject No.	Gender (M.F)	Age (Yrs)	Weight (Kg)	WBC (4.00–10.00) 10^3^/µL)	RBC (3.50–5.50) 10^6^/uL	HGB (11.0–16.0) g/dL	PLT (100–300) 10^3^/uL	Alkaline Phosphatase (53–128) U/L	Bilirubin Direct (0.00–0.20) mg/dL	Bilirubin Total (0.00–2.00) mg/dL	Total Protein (6.40–8.30) g/dL	Albumin (3.5–5.20) g/dL	AST/GOT (0.0–35.0) U/L	ALT/GPT (0.0–45.0) U/L	C-Reactive Protein (0.0–6.0) mg/dL	Ferritin (21.81–274.66) ng/mL
1	M	22	53	6.57	5.18	14	275	92	0.13	0.22	7.82	4.41	19.6	27.7	0.4	13.87
2	F	48	70.5	8.31	4.52	13.2	274	81	0.18	0.5	7.79	4.18	16.9	15.3	1.2	15.54
3	F	40	57.3	10.16	4.8	12	360	107	0.09	0.22	7.57	4.22	18.5	13.6	1.1	20.25
4	F	32	60	7.1	4.38	11.8	252	61	0.27	0.72	6.96	4.31	15.6	11.6	0.8	16.69
5	F	35	69	8.17	4.31	12.4	290	79	0.14	0.32	7.21	4.2	14.3	17.6	1.4	20.45
6	F	36	61	8.69	4.44	12	299	77	0.12	0.24	7.39	3.99	14.7	9.2	4.9	17.09
7	M	46	78	5.83	4.97	13.3	237	64	0.23	0.34	7.24	4.11	17.8	11.6	0.7	105.52
8	M	49	101.3	6.26	4.39	13.3	310	61	0.12	0.18	7.11	4.06	16.7	28.4	1.2	119.41
9	M	40	59.7	5.83	4.84	14.9	294	113	0.3	0.73	7.21	4.4	23.9	20.8	1.7	51.1
10	F	28	68	5.42	4.69	12.8	331	79	0.14	0.3	7.61	4.15	35.4	54.1	0.6	11.84
	Mean	36.59	66.67	7.09	4.64	12.94	290.21	79.68	0.16	0.34	7.39	4.20	18.64	18.21	1.10	26.84
	SD	8.80	13.89	1.54	0.29	0.98	36.15	18.05	0.07	0.20	0.29	0.14	6.29	13.38	1.29	40.32

**Table 2 medicines-10-00008-t002:** Basic laboratory parameters during the study course (*t*-Test, Paired two Sample for Means).

Sr No.	Blood Parameters	Baseline	Day 17	*p* Value
1	WBC (4.00–10.00) 10^3^/µL)	7.234	7.529	0.2801
2	RBC (3.50–5.50) 10^6^/uL	4.652	4.208	0.2834
3	HGB (11.0–16.0) g/dL	12.97	12.97	1
4	PLT (100–300)10^3^/uL	292.2	276.4	0.0724
5	Alkaline Phosphatase (53–128) U/L	81.4	80.4	0.5897
6	Bilirubin Direct (0.00–0.20) mg/dL	0.172	0.231	0.0994
7	Bilirubin Total (0.00–2.00) mg/dL	0.377	0.573	0.0658
8	Total Protein (6.40–8.30) g/dL	7.391	7.28	0.1193
9	Albumin (3.5–5.20) g/dL	4.203	4.255	0.1920
10	AST/GOT (0.0–35.0) U/L	19.34	17.09	0.1679
11	ALT/GPT (0.0–45.0) U/L	20.99	16.74	0.2398
12	C-Reactive Protein (0.0–6.0) mg/dL	1.4	1.61	0.4992
13	Ferritin (21.81–274.66) ng/mL	39.176	38.803	0.9009

**Table 3 medicines-10-00008-t003:** IL- 6 (0.00–7.00 pg/mL) changes after treatment in comparison with baseline, (RM-ANOVA test).

Test (Unit)	Visit	Statistics	*n* = 10	*p*-Value
**IL- 6 (0.00–7.00 pg/mL)**	Visit 1 (Baseline)	Mean (SD)	2.022 (0.794)	0.00094
		Min, Max	1.5, 3.7	
	Visit 2 (Day 17)	Mean (SD)	6.077 (2.845)	
		Min, Max	2.65, 12	
	Visit 3 (Day 34)	Mean (SD)	29.032 (12.774)	0.00008
		Min, Max	9.12, 44.6	
	Visit 4 (Day 60)	Mean (SD)	5.829 (6.771)	0.11934
		Min, Max	1.5, 24.26	

**Table 4 medicines-10-00008-t004:** CD panel in individuals during the study (*t*-Test, Paired two samples for Means).

Test (Unit)	Visit	Statistics	*n* = 10	*p*-Value
Absolute Lymphocyte count (990.00–3150.00)/uL	Visit 3 (Day 34)	Mean (SD)	2198.7 (526.201)	0.25587
		Min, Max	1563, 2854	
	Visit 4 (Day 60)	Mean (SD)	2287.9 (502.911)	
		Min, Max	1699, 3004	
Absolute CD4 (424.00–1509.00)/uL	Visit 3 (Day 34)	Mean (SD)	833.5 (280.553)	0.026799
		Min, Max	505, 1453	
	Visit 4 (Day 60)	Mean (SD)	900.8 (328.806)	
		Min, Max	526, 1520	
Absolute CD8 (169.00–955.00)/uL	Visit 3 (Day 34)	Mean (SD)	652.3 (272.316)	0.2373
		Min, Max	300, 1046	
	Visit 4 (Day 60)	Mean (SD)	687.7 (293.047)	
		Min, Max	351, 1153	

## Data Availability

Not applicable.

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
