# Peer review of "Safety and Evaluation of the Immune Response of Coronavirus Nosode (BiosimCovex) in Healthy Volunteers: A Preliminary Study Extending the Homeopathic Pathogenetic Trial"

_medicines, 2022, doi:10.3390/medicines10010008_

Round 1
Reviewer 1 Report
This manuscript describes a pilot study of a homeopathic remedy derived from COVID and designed to prophylactically reduce symptoms of SARS-CoV2 infection. The manuscript is well written and the study, while small, is strong.
There are several minor changes that could improve the manuscript.
1. The introduction is a strong description of the rationale for homeopathy and for pilot data. It would be improved by including the rationale for which immune measures were chosen. These statements are in the discussion and could simply be moved to the introduction.
2. Was the person from who the sample was obtained consented (line 84)?
3. Confirm that the human genetic material is destroyed in the process.
4. Get rid of this statement “Phase-1 studies are generally conducted using a 10-12 subject sample size.” (line 122) Pilot studies can be any size.
5. Missing Table 1 – Demographics. Need Age, Gender, BMI, etc of participants. Could include baseline measures (from current table 1) in this table.
6. Was CRP elevated at Day 34? Ah – found this in the discussion. Include the statement that CRP was not elevated in the results. IL-6 typically stimulates production of CRP. The fact that CRP is not elevated but IL-6 is interesting and should be discussed (currently one sentence in the discussion) – and this is a better reference for discussion of IL-6 and CRP. https://pubmed-ncbi-nlm-nih-gov.liboff.ohsu.edu/29499302/
Also, in future studies, measuring IL-6 in triplicate is recommended as this cytokine is responsive to multiple types of stimuli.
7. Table 3 – it would be helpful to have baseline CD4 and CD8 counts and P values – maybe split the data into two Tables, one for CD4 and one for CD8. Ah, again, this is addressed – in the last paragraph of the manuscript. Include in the results the sentence that “Lymphocyte count was measured after IL-6 data demonstrated results, therefore no baseline data is available.” Also mention that while lymphocytes increase in number, they remain within normal ranges, thus there is no fear that autoimmunity is triggered by this nosode.
Reviewer 2 Report
Major
1. Funding and ethical clearance statement must be provided. Conflict of interest should explicitly state whether the funders have any role in the study design and data interpretation.
2. This is not a controlled study. How authors justify this? Kindly state in the limitation.
3. Introduction. Background of COVID-19 is lacking. Kindly provide current limitations in COVID-19 management and prevention including: limited available approved drugs (doi: 10.52225/narra.v2i3.92) and ineffective vaccines again emerging SARS-CoV-2 variants (doi: 10.52225/narra.v2i3.88). Moreover, available anti-SARS-CoV-2, such as molnupiravir, also expose risk of toxicity to patients (doi: 10.1002/jmv.27730).
4. Methods not clear. How the “oropharyngeal swab” sample analyze? Using PCR?
5. “This nosode is prepared from a clinical sample as mentioned in N-IV group in the HPI [11]. Almost all the nosodes (such as Psorinum, Medorrhinum, Tuberculinum Carcinosin, Syphilinum) in the homeopathic literature and available in the market have been made from the clinical samples [12].” This is not COVID-19 nosode, why are they here?
6. “Ethics Committee constituted by Homeopathy India Private Limited” Provide the registration number
7. Standard deviation (SD) should be presented in the table.
8. Table 2 and 3 should just be in the supplementary file. The one presented in the manuscript should in the form of Mean±SD only along with the p-value from the post hoc test. The two table can be combined, please do so.
9. Discussion. Lengthy discussion about IL-6 and CD4 obscure the point of this study; kindly trim it short.
10. Discussion should be enriched with the emergence of SARS-CoV-2 variants (DOI: 10.1016/j.jiph.2022.11.024) to make the manuscript relevant to the current situation.
Minor:
1. Affiliation number 3 does not have complete address (City, Country).
2. Abstract. Use (:) instead of (.) for every section (Objectives, Methods, Results, Conclusions). Check again the guideline.
3. Keywords. ‘coronavirus’ should have initial capital letter.
4. Reveal the registration ID for clinical trial in the abstract.
5. The transition between paragraphs is bad. For example, the first and second paragraph of introduction. Please improve it.
6. “ml” should be “mL”
